# Effects of Two *Trichoderma* Strains on Plant Growth, Rhizosphere Soil Nutrients, and Fungal Community of *Pinus sylvestris* var. *mongolica* Annual Seedlings

**Saiyaremu Halifu [1], Xun Deng [2], Xiaoshuang Song [2] and Ruiqing Song [1,***

1   College of Forestry, Northeast Forestry University, Harbin 150040, China
2   Institute of Forestry Protection, Heilongjiang Forestry Academy, Harbin 150040, China
*   Correspondence: songrq1964@nefu.edu.cn; Tel.: +86-13804522836

**Abstract:** *Trichoderma* spp. are proposed as major plant growth-promoting fungi that widely exist in the natural environment. These strains have the abilities of rapid growth and reproduction and efficient transformation of soil nutrients. Moreover, they can change the plant rhizosphere soil environment and promote plant growth. *Pinus sylvestris* var. *mongolica* has the characteristics of strong drought resistance and fast growth and plays an important role in ecological construction and environmental restoration. The effects on the growth of annual seedlings, root structure, rhizosphere soil nutrients, enzyme activity, and fungal community structure of *P. sylvestris* var. *mongolica* were studied after inoculation with *Trichoderma harzianum* E15 and *Trichoderma virens* ZT05, separately. The results showed that after inoculation with *T. harzianum* E15 and *T. virens* ZT05, seedling biomass, root structure index, soil nutrients, and soil enzyme activity were significantly increased compared with the control ($p < 0.05$). There were significant differences in the effects of *T. harzianum* E15 and *T. virens* ZT05 inoculation on the growth and rhizosphere soil nutrient of *P. sylvestris* var. *mongolica* ($p < 0.05$). For the E15 treatment, the seedling height, ground diameter, and total biomass of seedlings were higher than that those of the ZT05 treatment, and the rhizosphere soil nutrient content and enzyme activity of the ZT05 treatment were higher than that of the E15 treatment. The results of alpha and beta diversity analyses showed that the fungi community structure of rhizosphere soil was significantly different ($p < 0.05$) among the three treatments (inoculated with *T. harzianum* E15, *T. virens* ZT05, and not inoculated with *Trichoderma*). Overall, *Trichoderma* inoculation was correlated with the change of rhizosphere soil nutrient content.

**Keywords:** *Trichoderma* spp.; growth promotion; *Pinus sylvestris* var. *mongolica*; soil microecological environment; high-throughput sequencing

## 1. Introduction

*Trichoderma* species belong to Hyphomyceteales, Hyphomycetes, Deuteromycotina, and Eumycota. Their existence in the natural ecological environment is widespread; for example, they are found in plant seeds, the rhizosphere, the phyllosphere, corms, and soils. These species have plant growth promotion and soil environment improvement abilities [1,2]. During the colonization in plant roots, the mycelia of *Trichoderma* fungi twine around the plant roots to form an appressorium-like structure, then penetrate the root epidermis layer and survive for a long time between the plant cells of the epidermis and the cortex [3], having a direct promotional effect on the growth of seedlings [4–6], nutrient uptake in the rhizosphere [7–10], and rhizosphere microbial community structure improvement [11–14]. Chang and Baler [15] treated seeds or roots of pepper, *Vinca*, chrysanthemum, tomato, and cucumber with a conidia suspension of *Trichoderma harzianum* T-203 and found that the germination rate of pepper

increased, the flowering period of *Vinca* came earlier, the number of chrysanthemum flowers increased, the plant height and fresh weight of all these plants increased. Furthermore, the dry weight of tomato, cucumber, and pepper fruits also increased significantly. During the interaction among *Trichoderma harzianum*, *Trichoderma virens*, and *Arabidopsis thaliana*, the contents of the JA (jasmonic acid) and (salicylic acid) SA and the number of lateral roots were all significantly increased [16,17].

　　Many nutrients in the soil exist in a sparingly soluble or insoluble state, which affects the circulation of nutrients in the soil to some extent. *Trichoderma* species promote nutrient uptake by secreting organic acids to dissolve minerals and activate nutrients in the soil, leading to the circulation and utilization of nutrients in the soil. At the same time, due to the strong colonization ability of *Trichoderma* species, they expand the contact area between the rhizosphere and soil and increase the secretion of extracellular enzymes such as sucrase, urease, and phosphatase, as well as organic acids in the rhizosphere to improve nutrient cycling and enzyme activity in the soil. Maeda [18] reported that *Trichoderma* species can decompose nitrogen compounds into available nitrogen and release less $NO_2$. Khan [19] and Harman [20] found that *Trichoderma* species convert nutrients into effective nutrients to increase soil nutrient circulation in the soil, enabling the reduction of the use of nitrogen fertilizer. Mbarki [21] found that *Trichoderma* inoculation increased the effective nutrient content and the soil enzyme activity to repair soil and promote plant growth. The imbalance of the soil microbial community structure is the main cause of soil-borne diseases, and its diversity is an important indicator to measure soil properties. Due to their advantages of fast growth and strong vitality, *Trichoderma* species rapidly occupy the growth space and absorb the nutrients needed. The *Trichoderma* genus also has the feature of hyperparasitism; it secretes cell wall-degrading enzymes such as chitinases, cellulases, xylanases, glucanases, and proteinases. *Trichoderma* species absorb nutrients through degrading soil microbial cells, leading to the change of the soil microbial community structure [22,23]. Wagner [24] and Yadav [25] found that *Trichoderma* inoculation increases nutrient content and microbial biomass in addition to improving the soil microbial community structure.

　　Mongolian pine (*Pinus sylvestris* var. *mongolica*), a geographical variety of Scots pine (*P. sylvestris*), is naturally distributed in the Daxinganling mountains of China (50°10′–53°33′ N, 121°11′–127°10′ E), in Honghuaerji of the Hulunbeier sandy plains of China (47°35′–48°36′ N, 118°58′–120°32′ E), and in parts of Russia and Mongolia (46°30′–53°59′ N, 118°00′–130°08′ E) [26]. It is often planted as an ornamental tree because of its height and greening characteristics. In addition, this tree is characterized by cold hardiness, drought tolerance, strong adaptability, and rapid growth [27,28]. It is currently the main coniferous tree species utilized in the"3-North Shelter Forest Program" and the "Sand-Control Project" in China, and plays an important role in ecological construction and environmental restoration.

　　The excessive and uncontrolled use of chemical fertilizers and pesticides have resulted in various adverse effects such as serious diseases, soil environmental damage, and poor growth of seedlings [29]. The utilization of beneficial microorganisms, including *Trichoderma* spp. and microbial metabolites, is a new environmentally friendly plant health management method compared with the use of chemical pesticides. This approach has the advantages of being pollution-free, residue-free, safe for natural enemies, difficult to produce resistance, and conducive to human and animal safety, as well as the advantage of environmental protection [30,31].

　　In this study, *T. harzianum* E15 (introduced from the University of Edinburgh, UK) and *T. virens* ZT05 (isolated from the Zhanggutai Experimental Forest Farm of Liaoning Province, China) were used to study the effects of *Trichoderma* spp. on the growth and root structure of annual seedlings of *P. sylvestris* var. *mongolica*, on the physical and chemical properties of rhizosphere soil, and on the fungi community structure. A comparison was made of effects of the introduced and local isolates of *Trichoderma* strains on seedling growth and the soil environment. The research objectives include the assessment of the effects of (1) *T. harzianum* strain E15 and *T. virens* strain ZT05 on annual seedling growth and root structure of *P. sylvestris* var. *mongolica*; (2) *T. harzianum* strain E15 and *T. virens* strain ZT05 on rhizosphere soil physicochemical properties and enzyme activities, which in turn affect the annual seedlings of *P. sylvestris* var. *mongolica*; and (3) *T. harzianum* strain E15 and *T. virens* strain ZT05

on the fungi community structure annual in the rhizosphere soil in which the *P. sylvestris* var. *Mongolica* seedlings were planted.

## 2. Materials and Methods

### 2.1. Organisms and Growth Conditions

Two *Trichoderma* strains were used in this research. *T. harzianum* E15 was introduced from the University of Edinburgh, UK to China. *T. virens* ZT05 was isolated from the rhizosphere soil of the *P. sylvestris* var. *mongolica* forest of the Zhanggutai Experimental Forest Farm of Liaoning Province (42°43′–42°51′ N, 121°53′–122°22′ E), China. These two strains were grown on a PDA medium (potato extract 12 g/L, dextrose 20 g/L, agar 14 g/L; Haibo Biotechnology, China) at pH 6.0. The *Trichoderma* species on PDA medium was cut with a sterile puncher ($\varnothing$ = 5 mm) after culturing for 5 days. Suspension cultures of the two *Trichoderma* strains were obtained by transferring the mycelium inoculum to liquid PD medium (PDA medium without agar) separately. Seven-day suspension cultures, maintained in the dark at 25 °C under agitation (150 rpm), were used to inoculate the seedlings [32].

The experiment seeds of *P. sylvestris* var. *mongolica* (purchased from the Zhanggutai Experimental Forest Farm in Zhangwu County, Liaoning Province, China) were surface-sterilized with potassium permanganate (0.5%, v/v) for 30 min, then washed five times with sterile distilled water. They were then germinated on sterile moistened gauze at 25 °C for 5 days. After germination, the seedlings were transferred to plastic pot (15 × 15 cm, 20 seeds per pot) filled with a sterile culture substrate—namely, a peat soil/vermiculite/sand (2:1:1, v/v/v) mixture that was sterilized in a high-temperature autoclave for 2 h at 121 °C. The pots were kept under greenhouse conditions (day/night thermal regime of 22/30 ± 3 °C, and 14 h light/10 h dark photoperiod) and watered every 2 days for 1 month, after which the seedlings were inoculated with the fungi [33,34].

### 2.2. Experimental Design and Seedling Inoculation

For all treatments, including the control, 20 pots (~20 seedlings per pot) were prepared, giving a total of 400 seedlings per treatment. There were three treatments: (1) inoculation with PD blank culture medium (CK); (2) single inoculation with *T. virens* ZT05; and (3) single inoculation with *T. harzianum* E15. The inoculations were performed by transferring 100 mL of the fungal suspension culture into the planting hole [35], where it was introduced at the root system level. The control plants were inoculated with 100 mL PD blank culture medium. All treatments were arranged at random under the greenhouse conditions given above.

### 2.3. Sampling and Analysis of Seedlings

The seedlings were harvested 3 months after inoculation. The harvest was conducted without damaging the root system, which was carefully washed to remove the soil. A total of 50 seedlings per treatment were randomly selected, the first 30 seedling were used to measure the biomass index. For each seedling, the biomass index included calculations of the plant height, ground diameter, fresh weight, and dry weight at harvest. Once the fresh weight had been measured, the seedlings were oven-dried at 85 °C for 5 h to measure the dry weight.

### 2.4. Soil Properties Analysis

Soil samples were collected from 100 randomly selected seedlings per treatment. Rhizosphere soil samples were collected from the root zone within 5 mm using a brush and passed through 1 mm mesh screen. Soil samples used to determine the enzyme activity and physicochemical properties were air-dried at 25 °C and collected into sterile sample bags, then kept in a 5 °C refrigerator until further assays.

Organic matter (OM) was measured using the potassium dichromate oxidation heating method [36]. Total nitrogen (TN) was determined using the Kjeldahl method [36], total phosphorus (TP) was

determined using Mo–Sb colorimetry [36], available phosphorus (AP) was determined using the antimony bismuth anti-colorimetric method with double acid leaching, rapidly available potassium (AK) was measured using a NH4OAc leaching flame photometer [36], and total potassium was determined using aflame photometer [36]. A pH meter [36] was used to determine soil pH (1:2.5). The soil saccharase, catalase, acid phosphatase, and urease activities were measured using a kit from Nanjing.

## 2.5. Fungal Diversity Analysis

The rhizosphere soil samples collected according to the method described above and 5.0 g soil samples per biological repetition were placed in 50 mL sterile centrifugal tube and transported to the laboratory in a cooler with an icepack. Soil samples used for high-throughput sequencing were stored in a centrifuge tube at −80 °C [37–39] until soil DNA extraction. For the high-throughput sequencing of soil microorganisms, the total genomic DNA was extracted from 0.5 g of soil using an EZNA Soil DNA Kit (Omega Bio-Tek, Norcross, GA, USA) according to the manufacturer's instructions. DNA was eluted with 100 µL of elution solution from the kit. The DNA sample concentration and quality (A260/A280 ratio) were measured using a NanoDrop2000 spectrophotometer (Thermo Scientific, Walthan, MA, United States). Each treatment had three replicates in our experiment. High-throughput sequencing analysis of the ITS region was performed to determine soil fungal communities. For each treatment, three replicates were sequenced. The primers ITS1 (5′-CTTGGTCATTTAGAGGAAGTAA-3′) and ITS2 (5′-GCTGCGTTCTTCATCGATGC-3′) were used to amplify the ITS1 region of the fungal ITS [10–13]. PCR was performed in a 20 µL reaction system: 4 µL of 5× FastPfu buffer, 2 µL of 2.5 m MdNTPs, 0.8 µL of each primer (5 µM), 0.4 µL of FastPfu polymerase, 0.2 µL of BSA, 10 ng of template DNA, and 11.6 µL of double-distilled water [37,38]. The PCR conditions were as follows: 95 °C for 3 min, 27 cycles of 30 s at 95 °C, 30 s at 55 °C, and 30 s at 72 °C, and with a final extension of 10 min at 72 °C. After PCR amplification, the obtained products were purified using an AxyPrep DNA Gel Extraction Kit (Axygen Biosciences, Union City, CA, USA) and quantified with QuantiFluor-ST (Promega, USA). Then, the purified amplicons were pooled in equimolar concentrations as a single aliquot and employed for library construction, and sequencing was performed on an Illumina MiSeq sequencer at Majorbio Biotechnology Co., Ltd. (Shanghai, China). Trimmomatic and FLASH software were employed to quality-filter and merge raw fastqfu files [39], while UPARSE software (version 7.1, http://drive5.com/uparse/) was employed to further analyze the pyrosequencing data. The sequences were then divided into operational taxonomic units (OTUs) with a 97% similarity cutoff, after which chimeras were removed using UCHIME [40,41]. RDP Classifier (http://rdp.cme.msu.edu/) was employed for the taxonomic annotation of each sequence within the confidence threshold of 0.7.

## 2.6. Data Analyses

Excel 2013 software was used for data processing. The differences of plant biomass, root structure index, soil pH, chemical properties, and soil enzymes were determined by one-way analysis of variance (ANOVA) in IBM SPSS 22.0 (IBM Corporation, New York, NY, USA). In addition, Pearson's method was performed for correlation analysis [37]. The differences were considered statistically significant at a 0.05 probability level in this study. Mothur software [42] was employed to analyze the alpha diversity index and rarefaction, and the coverage index was used to represent the sequencing depth index. The Ace (https://www.mothur.org/wiki/Ace) and Chao1 (https://www.mothur.org/wiki/Chao) indexes were used to represent the community abundance, while the Shannon (https://www.mothur.org/wiki/Shannon) and Simpson (https://www.mothur.org/wiki/Simpson) indexes were used to represent the species richness and diversity of the fungal community [43,44]. Additionally, the similarities analysis (ANOSIM) and heatmap analysis were calculated in the vegan package of R language [45,46] and the Unifrac distance calculation was performed for fungal beta diversity comparisons. The figures were generated using Origin 2019b (Origin Lab Corporation, Northampton, MA, USA).

## 3. Results and Analysis

### 3.1. Effects of Trichoderma Inoculation on Seedling Growth

#### 3.1.1. Seedling Height

*Trichoderma* inoculation promoted the growth of seedlings, as indicated by the significantly different heights between the control and treated groups ($p < 0.05$) (Figure 1). Specifically, the seedling height of the groups treated with ZT05 and E15 increased by 18.89% and 27.63%, respectively, relative to the control. Additionally, the height of seedlings treated with E15 was 7.35% greater than the height of those treated with ZT05.

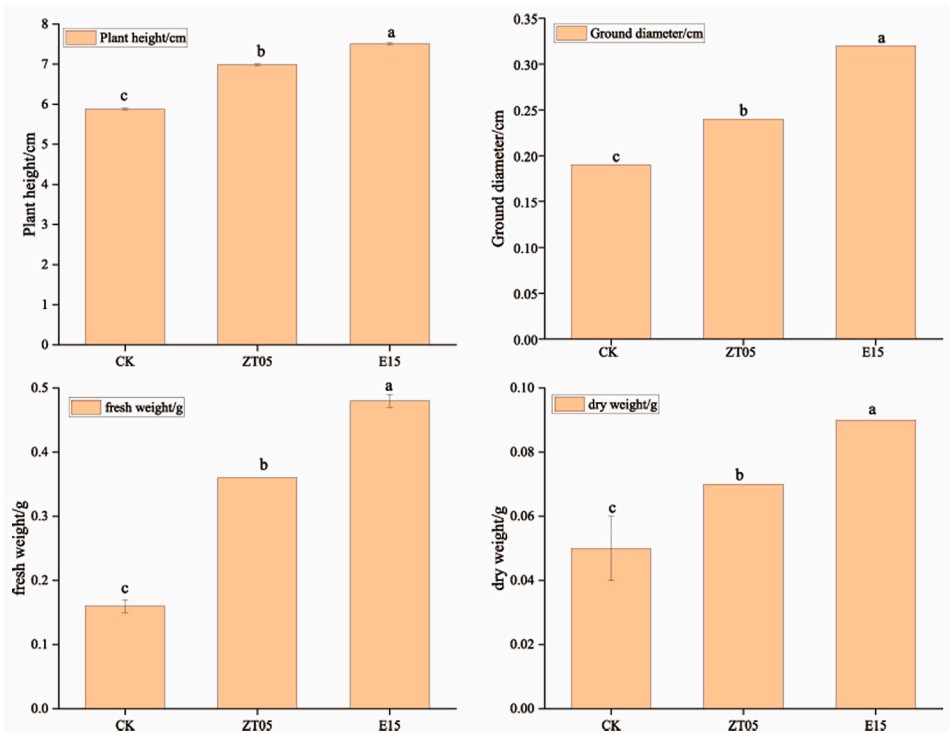

**Figure 1.** Effects of *Trichoderma* inoculation on plant growth. CK: inoculation with PD blank culture medium. ZT05: single inoculation with *T. virens* ZT05. E15: single inoculation with *T. harzianum* E15. Different letters indicate significant difference at $p < 0.05$, according to Duncan's new multiple range test. Vertical bars indicate standard error (SE).

#### 3.1.2. Seedling Diameter

As shown in Figure 1, the diameters of seedlings in the ZT05 and E15 treatment groups increased by 27.50% and 68.50%, respectively, relative to the control. Moreover, the E15 treatment group exhibited diameters increased 32.16% more than the ZT05 treatment group.

#### 3.1.3. Seedling Biomass

*Trichoderma* inoculation increased seedling biomass (Figure 1). The fresh weight of seedlings inoculated with ZT05 and E15 increased by 125.00% and 200.00%, respectively, compared to the control, and the seedling fresh weight of the E15 treatment group increased 33.33% more compared to the ZT05 treatment group. The dry weight of seedlings inoculated with ZT05 and E15 increased by 28.57% and 44.44%, respectively, compared to the control, and the seedling dry weight of the E15 treatment group increased 22.22% more compared to the ZT05 treatment group.

### 3.2. Effect of Trichoderma Inoculation on the Root Structure of Seedlings

Root length and surface area are important parameters for measuring the distribution of roots, while root average diameter, tip number, and branch number are important parameters for measuring root absorption efficiency. As shown in Table 1 and Figure 2, *Trichoderma* inoculation significantly increased root system parameters such as root length, root surface area, average root diameter, and number of root tips and branches ($p < 0.05$). When compared with the control, treatment with ZT05 increased root length by 25.11%, root surface area by 98.19%, average root diameter by 5.66%, root tip number by 45.89% and branch number by 74.42%. When compared with the control, treatment with E15 increased these parameters by 3.43%, 18.21%, 3.77%, 22.10%, and 31.40%, respectively. When compared with treatment with E15, these indexes were 20.96%, 67.66%, 1.82%, 19.48%, and 32.74% higher, respectively, following treatment with ZT05.

**Table 1.** Effects of *Trichoderma* inoculation on seedling root structure. CK: inoculation with PD blank culture medium. ZT05: single inoculation with *T. virens* ZT05. E15: single inoculation with *T. harzianum* E15. Different letters in the columns indicate significant differences ($p < 0.05$), according to Duncan's new multiple range test.

| Index | CK | ZT05 | E15 |
|---|---|---|---|
| Root length/cm | 68.14 ± 0.81 B | 85.25 ± 0.57 A | 70.48 ± 1.88 B |
| Surface area/cm$^2$ | 11.59 ± 0.10 C | 22.97 ± 0.72 A | 13.70 ± 0.31 B |
| Average diameter/mm | 0.53 ± 0.00 C | 0.56 ± 0.01 A | 0.55 ± 0.00 B |
| Apical number | 100.90 ± 2.13 C | 147.20 ± 0.99 A | 123.20 ± 2.09 B |
| Bifurcation number | 17.20 ± 0.36 C | 30.00 ± 0.63 A | 22.60 ± 0.83 B |

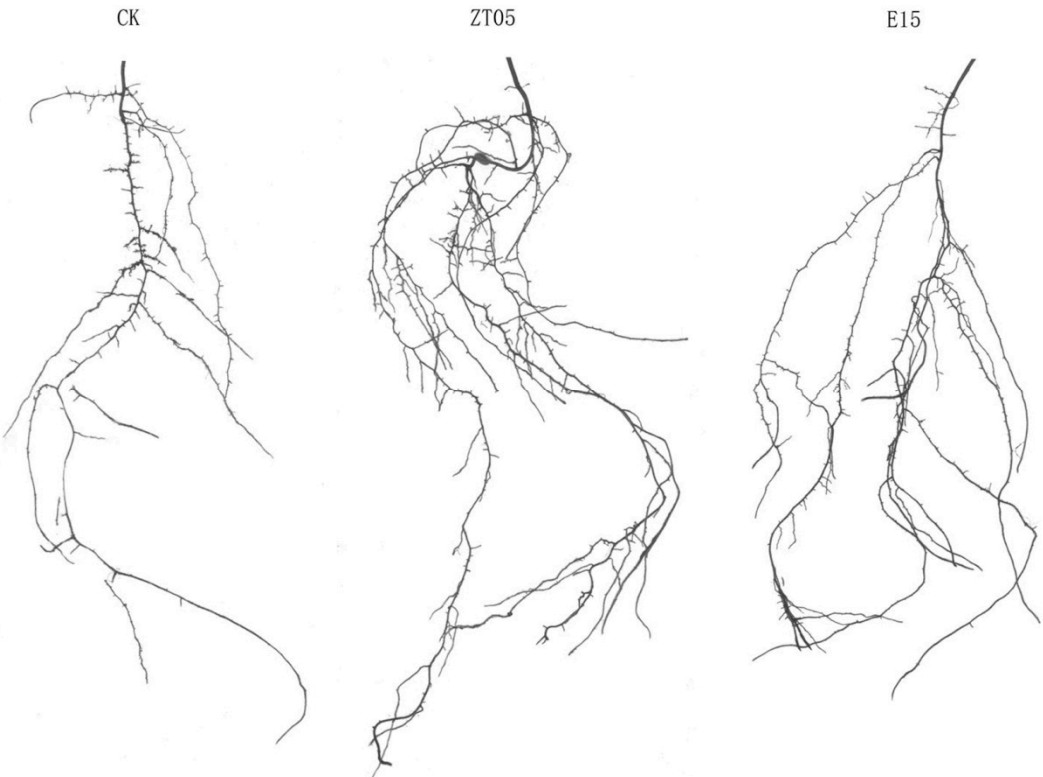

**Figure 2.** Effects of *Trichoderma* inoculation on seedling root structure of CK, ZT05, E15. CK: inoculation with PD blank culture medium. ZT05: single inoculation with *T. virens* ZT05. E15: single inoculation with *T. harzianum* E15.

### 3.3. Effects of Trichoderma Inoculation on Physicochemical Properties of Seedling Rhizosphere Soil

As shown in Table 2, significant differences were observed between the control and ZT05 and E15 treatment groups ($p < 0.05$). Specifically, the organic content of the control group was higher than that of the treatment groups inoculated with *Trichoderma*. This may have been due to the high capacity of *Trichoderma* to transform soil nutrients. Specifically, *Trichoderma* species can rapidly degrade nutrients produced by photosynthesis into a state in which they can be used for plant growth. The ability of *T. virens* ZT05 to transform soil nutrients was higher than that of *T. harzianum* E15. Trichoderma inoculation also significantly increased N and P nutrient contents in soil. This may be related to the ability of *Trichoderma* to degrade soil macromolecular nutrients into an effective state for plant utilization, thereby accelerating soil nutrient cycling and energy flow. However, organic matter refers to compounds of different compositions with primary components of C, N, and P. Total potassium levels showed no differences in CK, ZT05, and E15 groups ($p < 0.05$), while the levels of available potassium were organized in the order of CK > ZT05 > E15. These differences may have been a result of diversified microbial communities, slower plant growth, and smaller root systems in the control group than the two treatment groups. $CO_2$ released by root respiration as well as protons and organic acids secreted during the growth of root tip cells can lead to a change in pH. In the present study, the soil pH differed between the treated samples and the CK group ($p < 0.05$), with the pH values of samples treated with ZT05 and E15 increasing by 1.23% and 1.06%, respectively. This may be related to *Trichoderma* promoting plant growth by reducing plant respiration.

**Table 2.** Effects of *Trichoderma* inoculation on soil nutrients and soil enzyme activities of CK, ZT05, E15. CK: inoculation with PD blank culture medium. ZT05: single inoculation with *T. virens* ZT05. E15: single inoculation with *T. harzianum* E15. Different letters in the columns indicate significant differences ($p < 0.05$), according to Duncan's new multiple range test.

| Index | CK | ZT05 | E15 |
|---|---|---|---|
| OM g/kg | 84.27 ± 0.35 A | 69.77 ± 0.46 C | 77.91 ± 0.67 B |
| TN g/kg | 2.20 ± 0.01 C | 2.61 ± 0.00 A | 2.40 ± 0.01 B |
| AN mg/kg | 206.26 ± 0.03 B | 219.59 ± 0.59 A | 191.41 ± 0.36 C |
| TP g/kg | 1.87 ± 0.00 B | 1.77 ± 0.00 C | 1.95 ± 0.00 A |
| AP mg/kg | 772.14 ± 0.54 B | 796.76 ± 0.54 A | 459.50 ± 2.0 C |
| TK g/kg | 7.08 ± 0.07 A | 7.13 ± 0.09 A | 7.06 ± 0.12 A |
| AK mg/kg | 235.86 ± 0.26 A | 161.41 ± 0.48 B | 135.38 ± 0.07 C |
| pH value | 5.68 ± 0.00 B | 5.75 ± 0.01 A | 5.74 ± 0.01 A |
| Sucrase activity U/g | 4.27 ± 0.01 C | 20.82 ± 0.01 A | 14.45 ± 0.04 B |
| Catalase activity U/g | 5.15 ± 0.03 C | 5.97 ± 0.00 A | 5.57 ± 0.05 B |
| Acid phosphatase activity U/g | 3.81 ± 0.09 C | 8.78 ± 0.02 A | 5.06 ± 0.02 B |
| Urease activity U/g | 938.26 ± 0.08 C | 1295.74 ± 0.06 A | 1094.43 ± 2.42 B |

### 3.4. Effects of Trichoderma Inoculation on Rhizosphere Soil Enzyme Activity

Soil enzymes, which comprise the most active organic component in the soil biochemical process, are mainly derived from the secretions of soil microorganisms, plants, and animals. This component plays an important role in soil organic matter circulation and energy conversion. As shown in Table 2, *Trichoderma* inoculation led to a significant increase in the soil enzyme activity of the rhizosphere soil of seedlings. When compared with the control group, the sucrase, catalase, acid phosphatase, and urease activities of samples treated with ZT05 increased by 387.59%, 15.92%, 130.45%, and 38.02%, respectively. Sucrase activity, catalase activity, acid phosphatase activity, and urease activity of samples treated with E15 were increased by 238.41%, 8.16%, 32.81%, and 16.64%, respectively.

Sucrase activity, catalase activity, acid phosphatase activity, and urease activity in samples treated with ZT05 showed increases that were 44.08%, 7.18%, 42.36%, and 15.54% greater, respectively, than those seen in samples treated with E15. These results indicate that *Trichoderma* inoculation played an

important role in the circulation of nutrients and energy flow in soil, and strain ZT05 specifically had a significant effect on promoting soil enzyme and nutrient cycle activity.

*3.5. Effect on the Diversity of Rhizosphere Fungi of Seedlings*

3.5.1. Soil Sample Sequencing Results and Sampling Depth Verification

A total of 541,936 fungal sequences was obtained from nine mixed soil samples in three treatments using the Illumina MiSeqquome PE300 platform. Overall, 358 fungal OTUs were obtained upon OTU clustering at 97% similarity after separation and elimination. The unique fungal OTUs of CK, ZT05, and E15 groups amounted to 197, 17, and three, respectively. Overall, CK, ZT05, and E15 groups shared 53 OTUs, CK and ZT05 groups shared 62 fungal OTUs, and CK and E15 groups shared 13 fungal OTUs. These findings indicate that the fungal communities of the CK and ZT05 groups were more similar than the pairings of other groups (Figure 3).

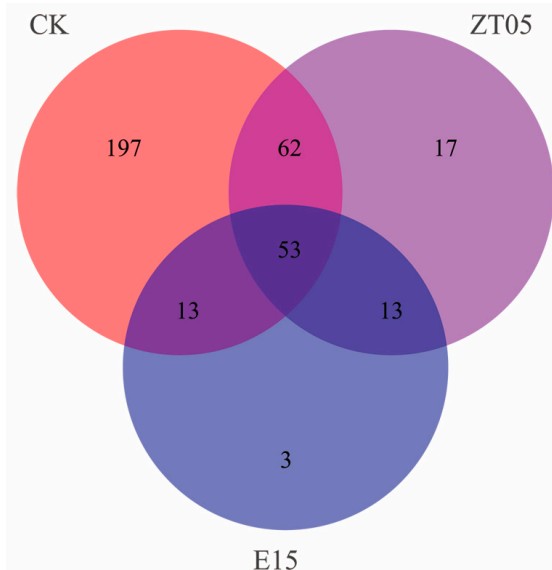

**Figure 3.** Venn diagram showing the shared operational taxonomic units (OTUs) of CK, ZT05, E15 treatments. CK: inoculation with PD blank culture medium. ZT05: single inoculation with *T. virens* ZT05. E15: single inoculation with *T. harzianum* E15.

3.5.2. Distribution of Soil Fungal Community

Categorical analysis of OTU representative sequences using a cutoff of 97% similarity revealed a total of seven phyla, 19 classes, 56 orders, 89 families, 11 genera, and 197 species of soil fungi. As shown in Figure 4, Ascomycota was the dominant fungi shared by the control, ZT05, and E15 groups (relative abundance ≥10%). In addition, Ascomycota was dominant in the control group (relative abundance 83.7%), while the relative abundances of Basidiomycota, Zygomycota, unclassified-k-fungi, and Chytridiomycota were 7.2%, 4.57%, 4.3%, and 0.086%, respectively. In the ZT05 treatment, Ascomycota was the dominant fungi (relative abundance 84.8%), while the relative abundances of Zygomycota, Chytridiomycota, unclassified-k-fungi, and Basidiomycota were 7.17%, 6.4%, 1.6%, and 0.04%, respectively. For the ZT05 treatment, Ascomycota was the dominant fungi (relative abundance 99.05%), while the relative abundances of Chytridiomycota, Zygomycota, unclassified-k-fungi, and Basidiomycota were 0.33%, 0.31%, 0.27%, and 0.03%, respectively. Significant differences were observed in the relevance abundances of Basidiomycota and unclassified-k-Fungi between the control and ZT05 and E15 treatments. The relevance abundance of Zygomycotain in the ZT05 group was significantly different from those of the control and E15 groups.

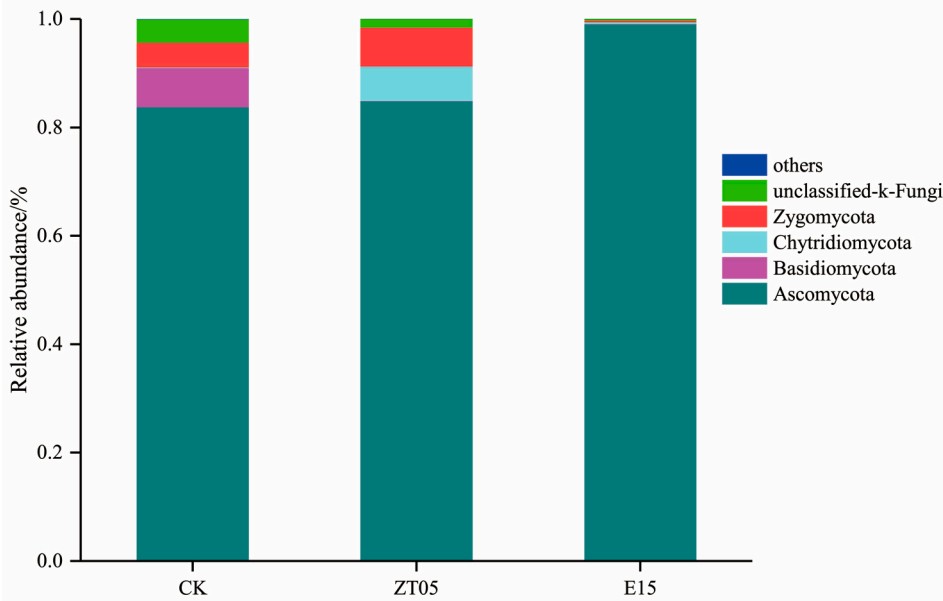

**Figure 4.** Relative abundance (%) of fungal phyla among CK, ZT05, E15 treatments, based on total sequence reads. CK: inoculation with PD blank culture medium. ZT05: single inoculation with *T. virens* ZT05. E15: single inoculation with *T. harzianum* E15.

The control group contained 134 genera of soil fungi, while the ZT05 group had 65 genera and the E15 group had 33 genera. Figure 5 shows the community analysis of the top 10 fungi at the genus level. In the control group, *Fusarium*, *Phoma*, and *Gibberella* were the dominant fungi genera (relative abundance%), while the relative abundances of *Trichoderma*, *Penicillium*, *Mortierella*, *Sphaerosporella*, *Rhizophlyctis*, unclassified-k-fungi, and *Monographella* were 1.09%, 5.16%, 1.72%, 7.31%, 0.08%, 4.34%, and 3.99%, respectively. For the ZT05 group, *Trichoderma* was the dominant fungi (relative abundance 76.40%), while the relative abundances of *Fusarium*, *Phoma*, *Gibberella*, *Penicillium*, *Mortierella*, *Sphaerosporella*, *Rhizophlyctis*, unclassified-k-fungi, and *Monographella* were 0.14%, 0.001%, 0.06%, 6.6%, 7.2%, 0.00%, 6.36%, 1.5%, and 0.0006%, respectively. For the E15 group, *Trichoderma* was the dominant fungi (relative abundance 98.41%), while the abundances of all the other genera were less than 0.33%.

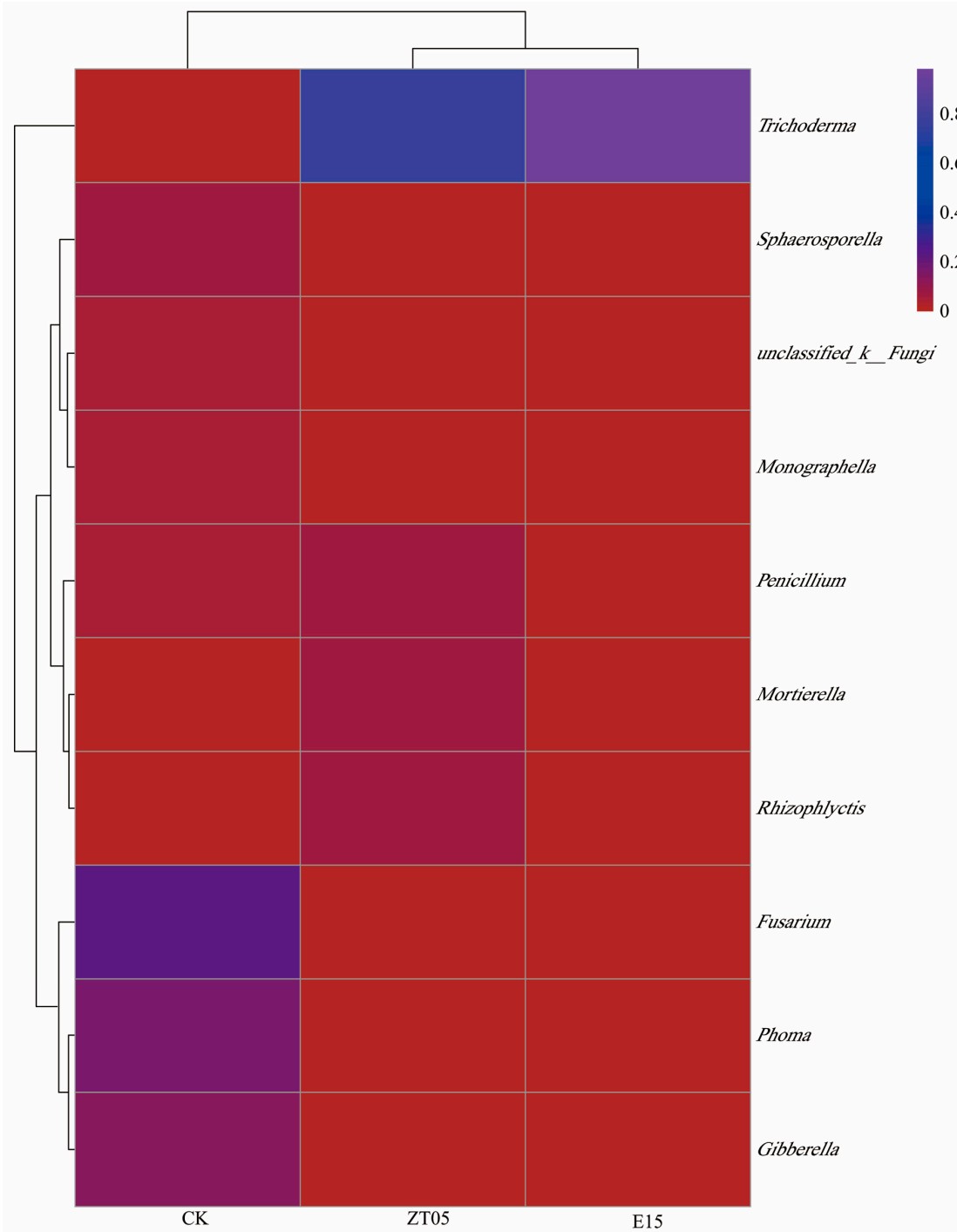

**Figure 5.** Heat map showed the relative abundance of the top 10 genes at CK, ZT05, E15 treatments. CK: inoculation with PD blank culture medium. ZT05: single inoculation with *T. virens* ZT05. E15: single inoculation with *T. harzianum* E15.

### 3.5.3. Analysis of α Diversity Index

ANOVA of the ITS rDNA diversity index of the CK, ZT05, and E15 soil samples was conducted. As shown in Table 3, the coverage indexes of the CK, ZT05, and E15 groups were close to 1, and they did not differ significantly. These findings indicate that the sequencing results could accurately reflect the actual situation of the tested soil samples. The Ace and Chao1 indexes of the soil samples were found to be in the order of CK > ZT05 > E15, with obvious differences observed between CK and the two treatment groups ($p < 0.05$). No significant difference was observed between the ZT05 and E15

groups, indicating that the total number and the richness of fungi in CK soil were higher than those in the two treatment groups. The total number and the richness of fungi communities in the ZT05 group were greater than those in the E15 group, but this difference was not significant. The Shannon index was found to be in the order of CK > ZT05 > E15, with significant differences observed between CK and the two treatment groups. No great difference was observed between the two treatments, indicating that the fungal richness of CK soil was higher than those of the other two groups. The Simpson index was found to be in the order of CK > E15 > ZT05, with significant differences observed between CK and the two treatment groups, but not between ZT05 and E15. These findings indicate that the complexity of CK soil was greater than that of the soil in ZT05 and E15 treatment groups; furthermore, the complexity of ZT05 soil was greater than that of E15 soil, but not significantly.

**Table 3.** Effects of *Trichoderma* inoculation on diversity indices of the soil fungal community in the CK, ZT05, E15. CK: inoculation with PD blank culture medium; ZT05: single inoculation with *T. virens* ZT05; E15: single inoculation with *T. harzianum* E15. Different letters in the columns indicate significant differences ($p < 0.05$), according to Duncan's new multiple range test.

| Samples | Shannon Index | Simpson Index | Chao1 Index | ACE Index | Coverage % |
|---|---|---|---|---|---|
| CK | 2.85 ± 0.44 A | 0.151 ± 0.07 A | 260.04 ± 16.39 A | 257.51 ± 17.23 A | 0.99 ± 0.00 A |
| ZT05 | 1.21 ± 0.15 B | 0.54 ± 0.08 B | 116.24 ± 3.43 B | 119.25 ± 3.71 B | 0.99 ± 0.00 A |
| E15 | 0.58 ± 0.02 B | 0.71 ± 0.01 B | 82.72 ± 5.71 B | 100.71 ± 12.45 B | 0.99 ± 0.00 A |
| *p* | 0.002 | 0.001 | 0.00 | 0.00 | 0.03 |

Pearson's analysis was used to analyze the correlation between the α diversity index and the physicochemical properties and enzyme activity (Table 4). The Ace, Chao1, and Shannon indexes were positively correlated with organic matter, available nitrogen, available phosphorus, and available potassium contents. In addition, theChao1 index was significantly positively correlated with the available potassium ($R^2 = 1.00$**, $p < 0.05$, ** indicates a very significant difference), while the Ace, Chao1, and Shannon indexes were negatively correlated with pH, total nitrogen, total phosphorus, sucrase activity, catalase activity, urease activity, and acid phosphatase activity. These findings indicate that *Trichoderma* has a crucial effect on soil nutrient cycling.

**Table 4.** Correlation analysis of diversity indices and soil properties. The correlation coefficient and significance were obtained using Pearson correlation analysis. Significant values are shown as: ** $p < 0.01$.

| | pH Value | Organic Matter | Available Nitrogen | Total Nitrogen | Available Phosphorus | Total Phosphorus | Available Potassium | Total Potassium | Sucrase Activity | Catalase Activity | Urease Activity | Acid Phosphatase Activity |
|---|---|---|---|---|---|---|---|---|---|---|---|---|
| Ace | −0.97 | 0.76 | 0.29 | −0.80 | 0.55 | −0.04 | 1.00 | −0.13 | −0.88 | −0.82 | −0.76 | −0.61 |
| Chao1 | −0.95 | 0.72 | 0.36 | −0.75 | 0.60 | −0.12 | 1.00** | −0.64 | −0.84 | −0.77 | −0.71 | −0.56 |
| Shannon | −0.92 | 0.65 | 0.49 | −0.69 | 0.68 | −0.21 | 1.00 | 0.03 | −0.79 | −0.71 | −0.64 | −0.47 |
| Simpson | 0.90 | −0.62 | −0.48 | 0.66 | −0.71 | 0.25 | −0.99 | −0.07 | 0.76 | 0.68 | 0.61 | 0.44 |

### 3.5.4. Analysis of β Diversity Index

The Bray–Curtis matrix was used to measure the heterogeneity of different sample communities in the soils. As shown in Figure 6, CK, ZT05, and E15 groups were distributed in different quadrants and the distribution distance was large, indicating that the composition of CK, ZT05, and E15 samples differed greatly. Nonparametric results were subjected to ANOSIM, which revealed that the differences between the fungal groups in the CK, ZT05, and E15 soil samples were greater than the within-group differences ($R = 1$), and that the differences were significant ($p = 0.003$).

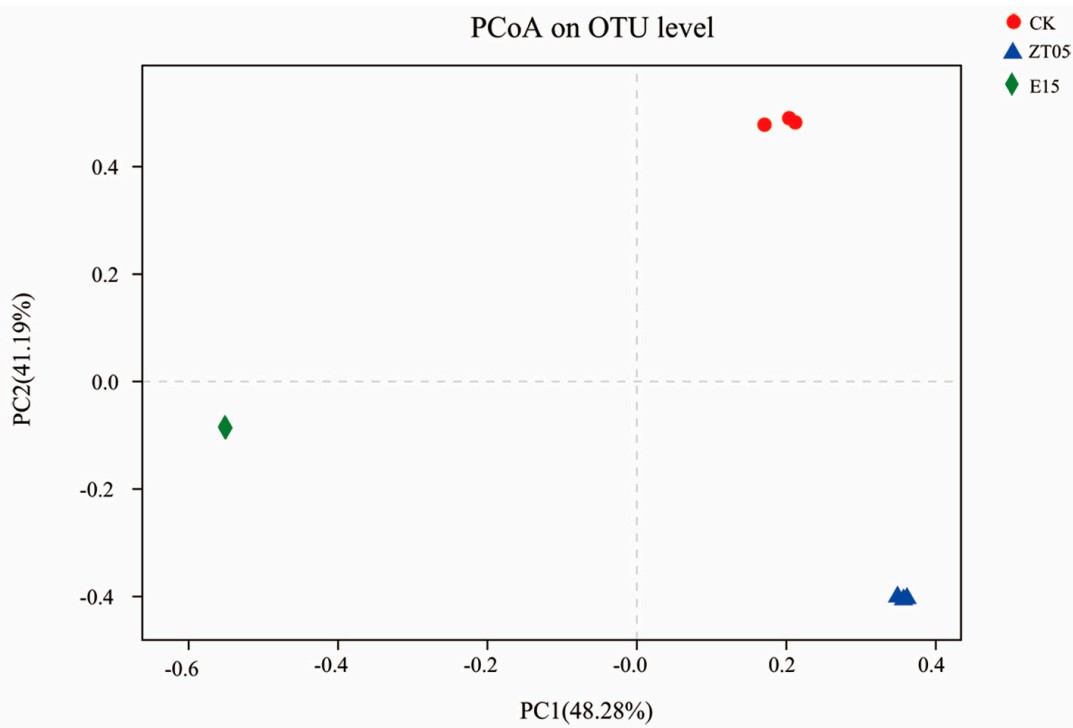

**Figure 6.** PCoA(Principal co-ordinates analysis) ordination based on Bray–Curtis similarities of fungal communities at CK, ZT05, E15 treatments. CK: inoculation with PD blank culture medium. ZT05: single inoculation with *T. virens* ZT05. E15: single inoculation with *T. harzianum* E15.

## 4. Discussion

*Trichoderma* spp. are soil fungi widely distributed in the natural environment which form symbioses with many plants [47]. The interaction between the plant rhizosphere and soil microorganisms plays an important role in plant growth, yield, nutrient cycling, and energy conversion in soil [48]. Plant root exudates promote the colonization of rhizosphere microorganisms, while soil microorganisms improve plant growth and increase the amount of nutrients in the soil environment by effectively utilizing plant photosynthates [49]. The beneficial effects of *Trichoderma* spp. can be divided into direct and indirect beneficial effects. The direct beneficial effects include promoting plant growth, promoting and improving plant root growth and structure [50], improving seed vigor and growth [51,52], and decomposing, recycling, and utilizing soil nutrients [4,53,54]. Our results show that inoculation with *T. harzianum* E15 and *T. virens* ZT05 could significantly promote seedling growth and change root structure in annual seedlings of *P. sylvestris* var. *mongolica.* The growth of seedlings inoculated with E15 was significantly higher than that of seedlings inoculated with ZT05, and the growth and structure of seedling roots inoculated with ZT05 was higher than that of seedling roots inoculated with E15.Many reports have shown that *Trichoderma* inoculation has significant promotion effects on plants seedlings and crops yields, such as those of cotton [54], tomato [55], and *Leymus chinensis* [8,30]. Harman [7] showed that the inoculation of *Trichoderma* spp. in maize could significantly promote plant growth, change root structure, and increase root activity. Shen [56], Fu [57], and Xiong [58] demonstrated that a *Trichoderma* agent stimulated banana root growth, promoted plant growth, and increased fruit yield. Hung [59] and Zhang [30] showed that a mixture of organic fertilizer and a *Trichoderma* agent could significantly improve plant growth and crop yield. *Trichoderma* could be used as an organic fertilizer as a growth substrate to degrade soil nutrients and improve the ability of plant photosynthesis, thereby improving plant growth. IAA is a molecule that is synthesized by plants and a few microbes [60]. In plants, IAA plays a key role in root and shoot development. The hormone moves from one part of the plant to another by specific transporter systems that involve auxin importer (AUX1) and efflux (PIN1-7) proteins. IAA is a key regulator of lateral root development

and root hair development [61]. Studies have shown that all *Trichoderma* spp. isolated from different geographical areas can secrete IAA and promote the growth of cucumber, bottle gourd, and bitter gourd [61]. Contreras-Cornejo [5] showed that IAA, a mycelial secretion of *Trichoderma* spp., could significantly improve plant and lateral root growth. The volatile and non-volatile secondary metabolites of *Trichoderma* spp.—including 6-n-pentyl-6H-pyran-2-one (6PP), gliotoxin, viridin, harzianopyridone, harziandione, and peptaibols [62,63]—have a significant growth-promoting effect on plants [7,64]. Vinale [65] showed that secondary metabolites of *T. harzianum* commercial strains T22 and T39, *T. atroviride* P1, and *T. harzianum* A6 also had significant growth-promoting effects on plant growth.

In this study, *Trichoderma* inoculation was found to significantly increase the nutrient content and soil enzyme activity in rhizosphere soil of *P. sylvestris* var. *mongolica* seedlings. *T. virens* ZT05 had a more significant effect on soil nutrient and enzyme activity in the rhizosphere of seedlings compared to *T. harzianum* E15. Many nutrient elements in soil exist in a slightly soluble or insoluble state, which limits the normal circulation of nutrients in soil. *Trichoderma* spp. can change the pH value of plants in the rhizosphere soil and secrete organic acids to degrade minerals such as large amounts of elements (P) and trace elements (Fe, Mn, and Zn), and activate soil nutrients, thus promoting the uptake of nutrients by plants as well as the recycling and utilization of nutrients in the soil environment [50,66,67]. At the same time, *Trichoderma* spp. have a strong colonization ability which, with the growth and extension of the root system, can increase the contact area between root and soil and increase the secretion of extracellular enzymes such as sucrase, urease, phosphatase, and organic acids in the rhizosphere, so as to improve the nutrient cycle and enzyme activity in the soil [6,68,69]. Yedidia [70] showed that under a hydroponic system, *T. harzianum* T203 could significantly increase the nutrient conversion and absorption of P, Fe, Mn, Zn, Cu, and Na, thus promoting cucumber growth and yield. Khan [19] showed that *Trichoderma* spp. can significantly improve the degradation and absorption of P, K, Ca, Mg, Cu, Fe, Mn, and Zn in fertilizers. Li [71] showed that the inoculation of tomato plants with *Trichoderma asperellum* CHF78 could significantly increase the soil's available nutrient content and plant nutrient uptake ability. Zhai showed that *T. asperellum* ACCC30536 improved the yield of *A. annua*, while the moisture, pH stability, organic matter content, and availability of nitrogen, phosphorus, and potassium in inoculated soil were also significantly improved [72]. El-Katatny found that *Trichoderma* spp. are important soil microorganisms that can have a significant effect on soil phosphorus, potassium, and nitrogen fixation, as well as on the restoration of degraded soil environments [67].

In this study, the Illumina Miseqquome PE300 high-throughput sequencing results show that Ascomycota was the dominant group in the rhizosphere soil of all treatments—CK, ZT05, and E15—and the relative abundance of Ascomycota in the E15 treatment rhizosphere soil was 99.05%. At the genus level, *Fusarium*, *Phoma*, and *Gibberella* were dominant in the CK treatment, while *Trichoderma* was the dominant genus in the ZT05 and E15 treatments. The results of the alpha and beta diversity analyses showed that *Trichoderma* spp. inoculation had a significant effect on the community structure of fungi in the rhizosphere soil of seedlings; this is consistent with the research results of Yu [73], Shen [60], Zhang [30]. *Trichoderma* spp. have the advantages of fast growth and vigorous vitality; thus, they can occupy the growing space quickly and absorb the required nutrients. At the same time, *Trichoderma* spp. can secrete cell wall-degrading enzymes including chitinases, cellulases, xylanases, glucanases, and proteinases, which can degrade microbial cells in the soil environment to absorb nutrients, thus changing the structure of the microbial community [22,23]. Stefania [74] showed that lemon plant soil microbial biomass increased by 46% after *Trichoderma* inoculation. Likewise, Mclean [75] showed that *Trichoderma* inoculation could change soil nutrient content and the soil microbial community structure of grassland soil.

## 5. Conclusions

1. *Trichoderma* inoculation increased the total biomass, seedling height, ground diameter, root length, root area, root diameter, number of root tips, and number of branches of *P. sylvestris* var. *mongolica* seedlings, thereby increasing the absorption area and growth potential of seedlings. The

contribution of *T. harzianum* E15 to seedling height, ground diameter, and total biomass was more significant. ZT05 had greater effects on root length, root area, root diameter, number of root tips, and bifurcation of seedlings.

2. *Trichoderma* inoculation increased soil nutrient content and soil enzyme activity in the rhizosphere soil of *P. sylvestris* var. *mongolica* annual seedlings. Specifically, *Trichoderma* inoculation had a significant effect on the community structure of fungi in the rhizosphere soil of seedlings, especially at the genus level.

**Author Contributions:** S.H., X.D., and R.S. conceived and designed the study. S.H., X.D., and Y.A. performed the experiments. S.H., X.D. and X.S. contributed to the sample measurement and data analysis. S.H. and X.D. wrote the paper.

**Funding:** This research was funded by [National Key Research and Development Program] grant number (2017YFD0600101), [National Natural Science Foundation of China] grant number (31670649, 31700564, 31170597, 31200484).

**Conflicts of Interest:** The authors declare no conflict of interest.

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
