# Peer review of "Effects of Two Trichoderma Strains on Plant Growth, Rhizosphere Soil Nutrients, and Fungal Community of Pinus sylvestris var. mongolica Annual Seedlings"

_forests, doi:10.3390/f10090758_

Round 1

Reviewer 1 Report

This study described the effects of two Trichoderma strains on soil fungal populations, soil physicochemical properties and on the growth of Pinus sylvestris seedlings.  The data largely corroborates the findings of many other related studies in other plant species.  

The English throughout the manuscript requires improvement in order to improve clarity and to avoid ambiguity.  

The title and abstract refer to stress resistance and yet there is no actual evidence to show that the seedlings are more resistant to stress.  The induction of putative stress-related responses alone is not proof enhanced stress resistance.  The last paragraph in the introduction alludes to biological control of plant disease by Trichoderma and yet there is no evidence of this in the manuscript.  However, an experiment to demonstrate suppression of damping off by these Trichoderma strains would add some substance to the inferred association.   

 Abstract and Introduction - English need improvement 

Materials and methods - line 96-7 - does not make sense. 

Line 111 - "...the pot with 30 seeds per pot..."  

114 - "...with Hoegel and solution..."  What is meant here?

127 - Trichoderma liquid? 

Results:

L209 - the reference to Table 1 should be Figure 1.   However, this data would be better if presented as a table.  Another option would be a bar chart.  Presenting the data as a line graph is not appropriate.   

L249 - change to glucanase.

L323-351 - numerous references to bacteria - incorrect.

Figure 4 - please converts to a Table.  

L353 - I couldn't find a reference to Figure 5 in the text.  The Figure caption requires more information to better describe the content. The Figure as presented is not particularly informative and the detail would be better conveyed if tabulated. I'm not sure if Figure 6 is required, this information could be presented as written text with the figure provided as supplementary information.  

Line 381 - Table 5 - only one of the parameters appears to show a significant correlation. Is it appropriate therefore to conclude in line 377 that Trichoderma had a crucial effect on soil nutrient recycling?     

Discussion - Many sentences should be re-written to improve clarity.  Most of the findings are not surprising as evidenced by the vast available literature on Trichoderma.  Greater effort is required to put the results of the current study into the context of the cited literature.  For this reason I would reduce the emphasis on stress and disease resistance and increase discussion on growth promotion, population structure and soil physicochemical properties.  Please try to describe what is new about this study and how this advances our understanding.  

Author Response

Thank you for your precious advice, and we have reviewed the paper according to your suggestions.Please see the attachment

Reviewer 2 Report

This is the review for the manuscript „Effects of two Trichoderma Strains on growth-promoting, stress-resistance and soil microecological environment of Pinus sylvestris var. mongolica seedlings” by Halifu et al.. Authors compared the effect of a control and two Trichoderma inoculations on tree fitness, root architecture, soil properties and microbial communities. Trichoderma is known to decrease the amount of harmful organisms and thus increase overall plant fitness. Results showed that indeed inoculation with Trichoderma affects microbial communities and accordingly plant fitness but also soil properties.
In it is current state the article is interesting but messy. The English is fine, but missing spaces hamper the reading. Besides, the given references are far from being complete. The presentation of methods and results is not ideal and can be highly improved.

The abstract is generally too detailed. It would be good to give first of all a greater context. What’s the motivation behind your study? Name this in one or two sentences and then you can go over to briefly mention your methods. Finally present your results but not as detailed. You can summarize soil properties or root architecture features and don’t need to name them individually. This will save you some space to introduce your study in more detail. Please also introduce that you have a control treatment. CK suddenly occurs without further mentioning. What do you understand as alpha-and beta-diversity?

For your keyword I suggest to be less repetitive. Please see: https://methodsblog.com/2015/12/18/seo/

The introduction lacks references. Many of you statements are not properly cited. Please correct this. Moreover, the construction of your arguments is not always sound. E.g. paragraph line 58-72. There you just write sentences, which are partly repetitive to previously mentioned things, but you also write like cause-effect structures without giving any cause or the effect is not fitting to the cause. This is just one example. Basically it accounts for your whole introduction. The flow is just not easy to follow. This is parly related to your rather unorthodox way to cite. Normally, if you use numbers, you start with one until X+1. But you ordered it alphabetically and used numbers. This makes it rather complicated. I would suggest you either count properly or you use the common way and write “Halifu et al., 2019”.

Line 41: A host plant has no hyphae. It’s called roots for plants.

Line 42: Here you mention pathogenic MOs and in the next sentence you refer only to mycelium. What about bacteria? Any what’s “mycelium that grows along pathogenic mycelium”?

The end of your introduction should lead to mentioning of your project, your research questions, your hypotheses and your methods to settle it in the greater framework. The introduction aims to unravel the research gap and your study eventually aims to close this mentioned gap. Thus, a general re-structuring of your introduction is highly necessary in order to publish it in Forests, which’s impact factor just increased.

The presentation of the methods is messy. You have many numbering, which I do not understand completely. I suggest the following order:

2.1. Experimental design: Where is the material from? How was it prepared? How was the pot experiment prepared? What were the conditions? (basically summarizing your 1.1.-1.4.

2.2. Characterization of Pinus sylvestris: All measures you carried out on the pine trees, including roots

2.3. Soil properties

2.4. Fungal diversity

2.5. Statistics

Why do you write only about fungi (line 172) but in Fig. 3 you refer to 16S? If you also analyzed bacteria, you have to mention it in your methods part. If this is a mistake and you studied only fungi, you should mention this as well and point this out already in your introduction.

The sections for fungal analyses and data analyses are partly overlapping. Try to separate the statistical analyses completely from the fungal section.

For the results, please also use a numbering which is common for publications in Forest.

Fig. 1: Points for individual treatments should not be connected by lines. This is simply wrong. There is no correlation between them.

Fig. 2: This is actually really nice and greatly depict the differences between treatments.

Fig. 4: fungal phyla names are not written in italics. Please correct your taxonomic affiliations. Zygomycota are no longer accepted.

Fig. 5: fungal genera are written in italics.

I have to admit that I checked the results not in detail. First of all you need to tidy-up the text in order to make it more untestable. In the next round, I will focus concretely on the result and discussion section. Please consider the hints I gave you for the introduction also for your discussion. Check carefully the construction of your arguments and give an outlook why your work was important. What is the greater good for the scientific community? Try to mention this and your paper will be highly cited. However, there is still a long way to go!

Author Response

Dear reviewer:

Thank you very much for review this paper, and give us lots of useful suggestion, and we have revised this paper according to the comments, and please review again. 

Thanks again.

Reviewer 3 Report

This paper discusses the effects of Trichoderma isolates on growth of Pinus sylvestrus seedlings. The results are of interest and of merit to the literature, but methods and results are not well presented. For example, there is no clear reference to the amount of replication in the plantings. Readers should be able to accurately replicate the study from the methods. It might help to have a colleague unfamiliar with the study read through and let you know what he could replicate it prior to submitting the manuscript for review.

Results could be better presented graphically. The use of a line graph is inappropriate for height and biomass. A bar graph would be best.

Conclusions drawn from this study are poorly supported. The study did not provide a stress challenge to the plants. The authors, therefore, cannot conclude that the Trichoderma induces stress tolerance, but MAY contribute to stress tolerance. The paper should be reworked to emphasize the growth and biomass results.

Author Response

(The authors gave the same response as above.)

Round 2

Reviewer 1 Report

Please take some time to correct the English through out the manuscript to improve clarity and to deliver the results more effectively.

Abstract: 

L14 - '..strong resistance.."  resistance to what?

L16-18 - this would read better if you you bring Trichoderma to the begining of the sentence e.g. The effects of Trichoderma E15 and ZTO5 on .."  

Introduction:

English need to be improved.  Be careful about which tense you are using.  

L175-181 - this text needs a reference.

L183 -  "..Trichoderma converted nutrients into effective nutrients.."  Poor description that lacks context.

L186-188 - reference required

L251-216 - poorly expressed

Materials and Methods: 

L603-606 - poor English

L1032-1038 - repeated text

Results:

Figure 5 is not particularly informative as a heat map and the data in the text are much more useful.  This data would be better and more useful to the reader if presented as a table.

Discussion:  please improve the English.

Conclusions:

delete 'can' from lines 1293 and 1309.

Reviewer 2 Report

Halifu and colleagues changed their manuscript. Particularly, the introduction and the discussion were completely rephrased. Unfortunately, the current version lacks proper scientific English and has still a big format issue. Many words are not properly separated by spaces from each other (e.g. line 321 “Chao1andShannonindexes” – these are four words and not one). I mentioned this already in my first review and still find the reading rather disturbing. Moreover, references are still missing, e.g. for PCR primers and statements within the introduction and discussion.

With respect to my time, I will not give a detailed judgment for this paper as it is currently. I ask the authors to correct their English, maybe even suggest a professional language service. I guess, it is just a matter of time and concentration to improve the formal presentation. I believe that the scientific outcome of your study can only benefit from these actions.

Reviewer 3 Report

The authors have done an excellent job revising this paper for publication. They have addressed major concerns regarding presentation of methodology and over-emphasis of the stress tolerance hypothesis. The graphics are also clearer.

There is still a little minor English and punctuation editing, but the paper is almost ready for publication.
